# Deconstructing What Makes a Good Optimizer for Autoregressive Language Models

**Rosie Zhao**[*]
Harvard University

**Depen Morwani**[*]
Harvard University

**David Brandfonbrener**[*]
Kempner Institute at Harvard University

**Nikhil Vyas**[*]
Harvard University

**Sham Kakade**
Kempner Institute at Harvard University

## Abstract

Training language models becomes increasingly expensive with scale, prompting numerous attempts to improve optimization efficiency. Despite these efforts, the Adam optimizer remains the most widely used, due to a prevailing view that it is the most effective approach. We aim to compare several optimization algorithms, including SGD, Adafactor, Adam, Lion, and Sophia in the context of autoregressive language modeling across a range of model sizes, hyperparameters, and architecture variants. Our findings indicate that, except for SGD, these algorithms all perform comparably both in their optimal performance and also in terms of how they fare across a wide range of hyperparameter choices. Our results suggest to practitioners that the choice of optimizer can be guided by practical considerations like memory constraints and ease of implementation, as no single algorithm emerged as a clear winner in terms of performance or stability to hyperparameter misspecification. Given our findings, we further dissect these approaches, examining two simplified versions of Adam: a) signed momentum (Signum) which we see recovers both the performance and hyperparameter stability of Adam and b) Adalayer, a layerwise variant of Adam which we introduce to study the impact on Adam's preconditioning for different layers of the network. Examining Adalayer leads us to the conclusion that, perhaps surprisingly, adaptivity on *both* the last layer and LayerNorm parameters in particular are necessary for retaining performance and stability to learning rate.

## 1 Introduction

As language model architectures increase in scale, pretraining becomes more expensive. In response, numerous efforts have been made to design efficient optimizers to mitigate these costs, and yet Adam (Kingma & Ba, 2015) remains the primary optimizer used for training language models. This persistent preference for Adam is rooted in an underlying belief that Adam generally outperforms alternative optimization algorithms. Although newly proposed optimizers run ablations to demonstrate superior performance to Adam for select architectures and tasks (Liu et al., 2024; Chen et al., 2023), there is no consensus among the literature about the relative performance of these optimizers. In fact, to the best of our knowledge, Kaddour et al. (2024) is the only work comparing these optimizers but in the context of masked language modeling and at a single model scale.

In this work, we aim to rigorously evaluate whether the widespread reliance on Adam is justified and to further explore the characteristics that make adaptive optimizers particularly effective for training autoregressive language models. As such, we perform a comprehensive sweep across different optimizers, hyperparameters, architectures, and scale. Along with looking at optimal performance, we argue that due to the difficulty of hyperparameter tuning with increasing scale (Yang et al., 2021), the *stability* of performance with respect to hyperparameter choices is equally important. Prior work has explored the learning rate stability of Adam (Wortsman et al., 2024). We extend this investigation to include the stability of multiple optimizers with respect to various hyperparameter

---

[*]Equal contribution, randomized author ordering. Correspondence to `rosiezhao@g.harvard.edu`.

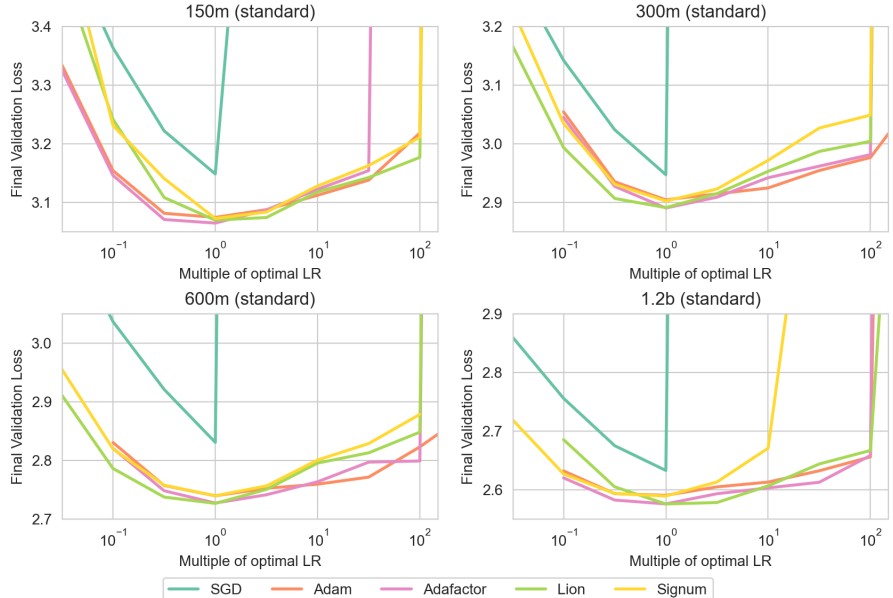

Figure 1: Final validation loss when training language models with 150m, 300m, 600m, and 1.2b parameters, sweeping across learning rates for five standard optimizers (SGD, Adam, Adafactor[1], Lion, and Signum). Plots have been shifted to align the optimal learning rates for each optimizer. Except for SGD, other optimizers seem comparable in their optimal performance and stability with respect to learning rate tuning.

choices. Surprisingly, we find that multiple optimizers introduced in the literature after Adam—such as Lion (Chen et al., 2023) and Adafactor (with momentum) (Shazeer & Stern, 2018; Zhai et al., 2022)—demonstrate robustness comparable to Adam and significantly superior to SGD. Figure 1 illustrates the remarkable similarity in performance and robustness of these optimizers across different learning rates and across multiple model scales (150m, 300m, 600m, and 1.2b parameters). This challenges the prevailing notion that Adam should be the default optimizer, where we see no single algorithm emerged as a clear winner in terms of performance or hyperparameter stability.

Following our initial ablations, we wish to identify the essential components of these optimizers that facilitate performance and stability. Thus, we conduct a series of investigations of simplified versions of these algorithms. We study signed momentum (Signum) (Bernstein et al., 2018; 2019), a special case of Lion. Prior works have also studied its similarities to Adam (Balles & Hennig, 2018). We find that Signum also recovers the stability and performance exhibited by Adam. This finding aligns with recent work (Kunstner et al., 2023), suggesting that the primary distinction between SGD and Adam is driven by Adam's resemblance to signSGD.

To further understand the role of preconditioning on various network parameters, we study Adalayer, which performs preconditioning on a *per-layer* basis. We empirically demonstrate that this variant nearly recovers the stability and performance of the other optimizers in previous ablations. Through empirical studies of Adalayer and its variants, we show that adapting the parameters of the last layer and LayerNorm parameters in a transformer is necessary to achieve stability and performance.

**Main contributions**

- We empirically study the stability to hyperparameters of various optimization algorithms including SGD, Adam, Lion and Adafactor, showing that with the exception of SGD, these optimizers are comparable in terms of both performance and hyperparameter stability. This holds across multiple scales (150m, 300m, 600m, 1.2b) and across two transformer architecture variants (Section 2).

---

[1]Our implementation of Adafactor adds back momentum as described in Section 2.2.

- To dive into the reasons behind the stability and performance of these algorithms, we empirically examine signed momentum (Signum), and show that it recovers their stability and performance (Section 2.6).

- We study a coarser variant of Adam called Adalayer, that does per-layer preconditioning and recovers much of the stability and performance exhibited by Adam (Section 3.1).

- Through an empirical study of Adalayer and its variants, we study the importance of adaptivity with respect to different layers of the network. (Section 3.2).

## 2 COMPARING OPTIMIZERS ACROSS HYPERPARAMETERS, ARCHITECTURES AND SCALE

### 2.1 METHODOLOGY

To conduct our experiments, we start with hyperparameters recommended by previous work (e.g., $\beta_1 = 0.9$). We initially perform a learning rate sweep to identify the optimal learning rate. After determining the optimal learning rate for each algorithm, we conduct one-dimensional sweeps for each of the other hyperparameters.

A limitation of this methodology is the potential neglect of "higher-dimensional" interactions between hyperparameters. This is an important direction for future work, but beyond the computational budget of this project. For example, some parameters like batch size and learning rate indeed are likely to exhibit 2D interactions (Shallue et al., 2019; Porian et al., 2024). However, we argue that the 1D sweeps provide a tractable methodology that gives us useful signal about the hyperparameter stability of a variety of algorithms around the parameters that are common in practice.

### 2.2 SETUP

We train decoder-only language models on C4 tokenized with the T5 tokenizer (Raffel et al., 2020) and report results in terms of validation loss. Due to the computational resources required to sweep over numerous hyperparameters, optimizers, and scales, we restrict the scope of our work to autoregressive language models, leaving other domains and language model architectures as future work. As we discussed in the introduction, we argue that it is best to evaluate algorithms both in terms of the loss achieved by the best hyperparameters (performance) as well as the robustness across values of the hyperparameters (stability).

In this section we first present the results of sweeps across the two most sensitive hyperparameters: learning rate and momentum ($\beta_1$). We sweep across five algorithms: Adam, Adafactor, Lion, Signum, and SGD. Further ablations of weight decay, warmup, $\beta_2$, and $\epsilon$ for the 150m standard model can be found in Section 2.5. Before diving into the results, we provide some more details about the setup.

**Algorithms.** We use the standard Pytorch implementation of AdamW (Paszke et al., 2019), the timm implementation of SGDW (Wightman, 2019), and the OLMo implementation of Lion (Groeneveld et al., 2024). Following Zhai et al. (2022) we implement ourselves a modified version of Adafactor which maintains the factored estimates of second moments but **has momentum** i.e. it is equivalent to Adam with factored second moment estimates. Since Signum is equivalent to Lion with $\beta_1 = \beta_2$ we reuse the OLMo implementation of Lion (Groeneveld et al., 2024) for it. We conducted experiments with the Sophia optimizer (Liu et al., 2024) in Appendix G. However, since it does not outperform Signum (which can be achieved by setting $\rho = 0$ in Sophia), we did not include it in other plots.

**Models.** We start from the OLMo codebase (Groeneveld et al., 2024) and train decoder-only transformer models of four sizes: 150m, 300m, 600m, and 1.2b, where the parameter count refers to non-embedding parameters. The models have widths of 1024, 1024, and 1408 and depths of 12, 24, 24. The MLP hidden dimension is 4x of the width. The activation function is GeLU (Hendrycks & Gimpel, 2016). We use RoPE positional encodings (Su et al., 2024). Attention heads are always dimension 64. We use PyTorch default LayerNorm. Following Wortsman et al. (2024) we do not learn biases for the linear layers or LayerNorms. We train in mixed precision with bfloat16.

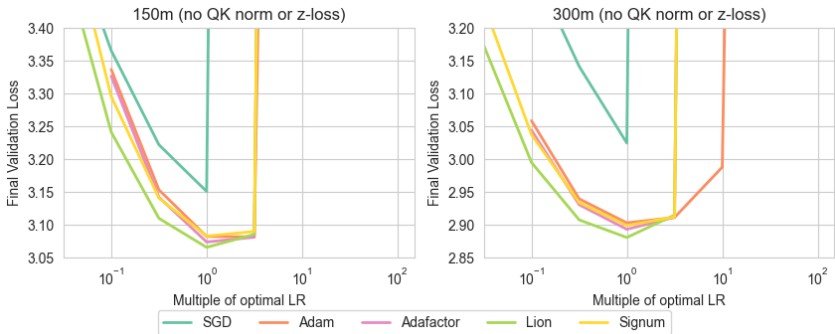

Figure 2: Sweeping learning rate without QK norm or z-loss for (**Left**) the 150m model, and (**Right**) the 300m model. These models are less stable than the standard model, but the same general trend across algorithms hold here.

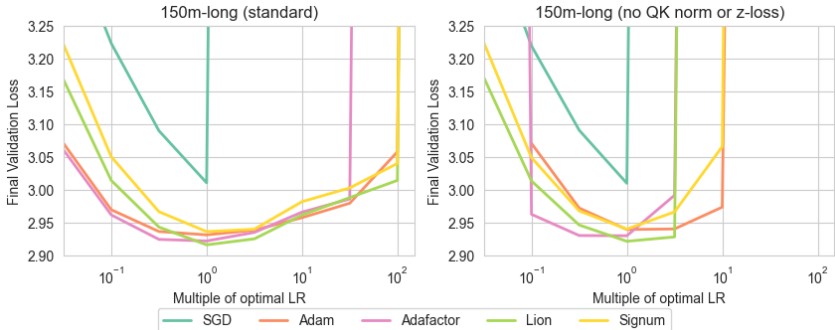

Figure 3: Sweeping learning rate on 150m models trained for 4x longer (100k steps) than in the base runs for (**Left**) the standard model, and (**Right**) the model without QK norm or z-loss. Compared to the shorter runs, these models achieve better performance and increased stability across learning rates.

**Training variants.** We note that Wortsman et al. (2024) observe that QK LayerNorm (Dehghani et al., 2023) and z-loss (Chowdhery et al., 2023) can have substantial effects on the stability of model training. As such, we consider two variants in our experiments: **standard** which refers to a model with QK LayerNorms and z-loss with coefficient 1e-4, and **no QK norm or z-loss** which refers to the same model without the QK norm layers or the z-loss.

**Token counts.** For all models, we use a batch size of 256 and sequence length of 512 (as in Wortsman et al. (2024)). We default to training models for the approximately "chinchilla optimal" number of tokens that is ≈20 times the number of parameters. Explicitly, this means for the 150m models we train for 25k steps or ≈3.3b tokens. The 300m models are trained for 50k steps, the 600m models are trained for 100k steps, the 1.2b models are trained for 200k steps, and the 150m-long models are also trained for 100k steps.

**Other hyperparameters.** We default to using 0 weight decay. We default to using a learning rate schedule with 10% of the training steps for warmup and then cosine decay with a minimum that is 10% of the maximum learning rate. We default to $\beta_2 = 0.95$ and $\epsilon = 1e\text{-}15$ following Wortsman et al. (2024). These parameters are ablated in Section 2.5.

## 2.3 SWEEPING LEARNING RATES

First, we sweep over the most important hyperparameter: learning rate. Note, in all of these sweeps over learning rate we set $\beta_1 = 0.9$ for all algorithms except for SGD, where we set $\beta_1 = 0.98$. As

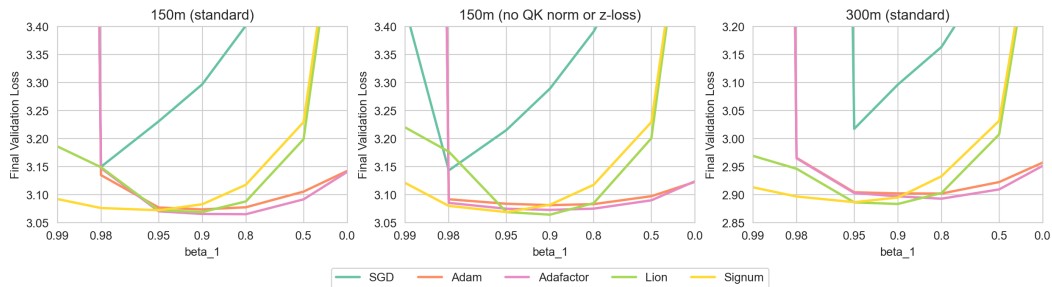

Figure 4: Sweeping momentum for fixed learning rate across three settings: (**Left**) 150m standard, (**Middle**) 150m with no QK norm or z-loss, (**Right**) 300m standard. Adam and Adafactor are similarly robust to $\beta_1$, while Lion and Signum are slightly more sensitive to low values and SGD is substantially more sensitive.

we will see in the following subsection, SGD is more sensitive to the momentum hyperparameters and requires more momentum to be competitive with the other optimizers.

Main results for our standard architecture across four scales are presented in Figure 1. Note that the x-axis shifts the learning rates to align the optimal learning rates across algorithms. In terms of absolute learning rates, we sweep in multiples of $\sqrt{10}$ from 1e-4 to 1 for Adam and Adafactor, from 1e-5 to 1e-1 for Lion and Signum, and from 1e-3 to 10 for SGD. We report on the optimal learning rates for each optimizer in Appendix B.

The key takeaway is that not only do the algorithms achieve similar performance at the optimal learning rate, but the learning rate stability itself is similar across algorithms and scales. The one exception is SGD, which is worse both in terms of optimal performance and in terms of stability.

Further ablations are presented in Figure 2 and Figure 3 illustrating performance for models with no QK norm or z-loss and 4x longer training time respectively. While we find that the architecture choices can clearly impact the amount of stability to learning rate, the cross-algorithm comparisons remain the same: Adafactor and Lion are competitive with Adam, while SGD is worse both in terms of performance and stability to learning rate. Similarly, training for longer can improve performance and stability to learning rate, but does not change the high-level cross-algorithm comparisons.

**Takeaway:** performance and stability to learning rate are comparable across the non-SGD algorithms that we tested.

### 2.4 SWEEPING MOMENTUM

Now we also sweep across momentum values (i.e. $\beta_1$)[2]. To do this sweep we fix the per-algorithm learning rate to be the optimal learning rate from the corresponding learning rate sweep.

Results are presented in Figure 4. We observe that across various settings, the robustness to $\beta_1$ is similar across the non-SGD algorithms when we stay in the range of momentums between 0.8 and 0.98. However, for high $\beta_1$ Lion is better and low $\beta_1$ Adam and Adafactor are better. Again we observe SGD being very sensitive to momentum.

**Takeaway:** performance and stability to momentum are comparable across the non-SGD algorithms that we tested if we stay within the usual range of momentum values.

### 2.5 ADDITIONAL HYPERPARAMETER SWEEPS

We also sweep over a variety of other hyperparameters in Figure 5 using the best per-algorithm learning rate and momentum. We observe that SGD is less stable with respect to weight decay and

---

[2]Note that in Lion, both $\beta_1$ and $\beta_2$ can be thought of as different types of "momentum" with $\beta_1$ being the "one-step" momentum and $\beta_2$ the "long-term" momentum. For consistency, we only sweep $\beta_1$ here and sweep $\beta_2$ in Section 2.5.

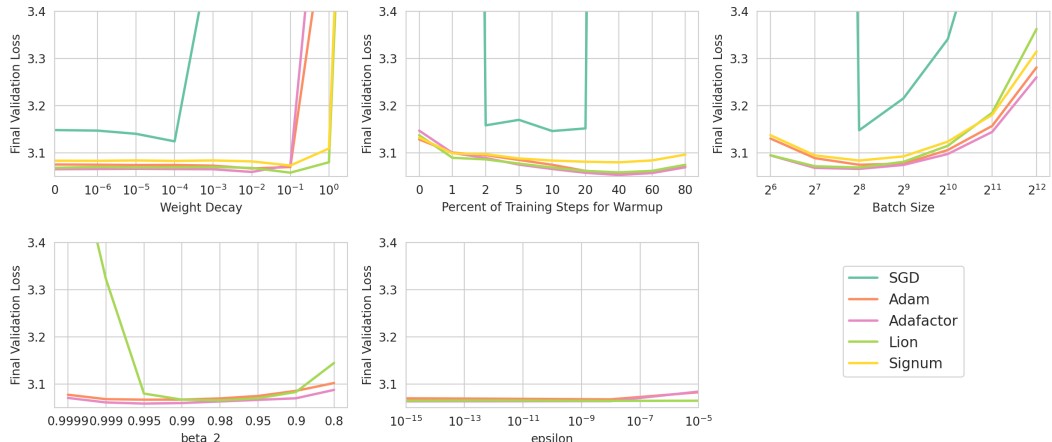

Figure 5: Sweeps over other hyperparameters. Top: weight decay, warmup duration, and batch size. Bottom: $\epsilon$ and $\beta_2$. We generally find little effect for the non-SGD algorithms, however there are parameters that differ from our defaults that can offer up to 0.02 improvements in validation loss.

warmup length. And while it is possible to get small benefits from higher weight decay, longer warmup, and higher $\beta_2$ than our defaults, generally the algorithms are much more stable to these parameters than learning rate and momentum. We observe two exceptions: Lion shows poorer performance at extreme values of $\beta_2$. However, it is important to note that the $\beta_2$ parameter in Lion functions more like a form of "momentum", whereas in Adam and Adafactor, $\beta_2$ regulates the moving average of the squared gradients. We also see that the performance across optimizers worsens at high batch size. This is expected as optimization algorithms are known to exhibit diminishing performance with increasing batch size (Shallue et al., 2019; Zhang et al., 2024a).

**Takeaway:** generally algorithms are more stable with respect to other hyperparameters— with the exception of batch size— and the possible gains in performance are relatively small compared to learning rate and momentum.

## 2.6 SIGNUM RECOVERS THE PERFORMANCE AND STABILITY OF ADAM

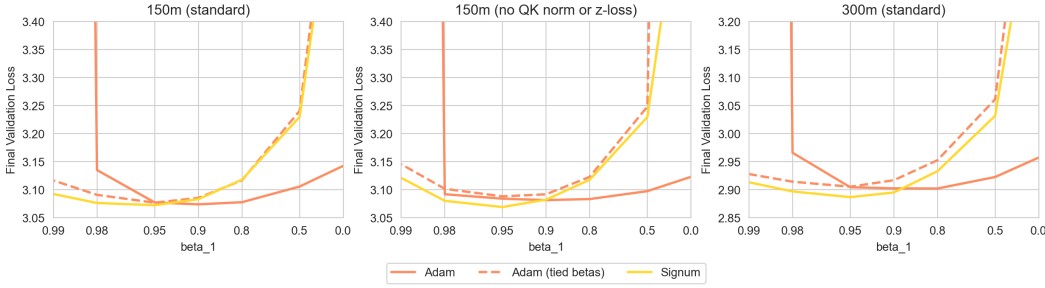

Figure 6: Sweeping momentum with $\beta_1 = \beta_2$ tied together for Adam (dashed) and compared to Signum and Adam with fixed $\beta_2 = 0.95$ (solid) across three settings: (**Left**) 150m standard, (**Middle**) 150m with no QK norm or z-loss, (**Right**) 300m standard. When $\beta_1 = \beta_2$, Adam behaves very similarly to Signum.

In Figure 1 we observed that Adam and Signum have similar performance and stability for language modeling, even at scale. The following lemma from prior work(Balles & Hennig, 2018) shows that Adam performs variance-adjusted sign gradient descent.

**Lemma 1** ( Balles & Hennig (2018)). *Consider a parameter with a history of gradients $g_t, g_{t-1}, \ldots$. Let $m$ be the random variable that is equal to $g_{t-\tau}$ with probability $(1 - \beta_1)\beta_1^\tau$ and $v$ be the random*

*variable that is equal to $g_{t-\tau}$ with probability $(1 - \beta_2)\beta_2^\tau$. The Adam update $\delta_{Adam}$ and the Signum update $\delta_{Signum}$ are related by*

$$\delta_{Adam} = \delta_{Signum} \cdot \frac{|\mathbb{E}[m]|}{\sqrt{\mathbb{E}[v^2]}}$$

If $\beta_1 = \beta_2$ then $m = v$ in Lemma 1 and hence the parameter-wise ratio of Adam and Signum updates is equal to the ratio of the mean and the square root of second moment of $m$. Intuitively, this holds because when $\beta_1 = \beta_2$, the first moment estimates of Signum and Adam, and second moment estimates of Adam, average the previous gradients with same coefficients $((1 - \beta)\beta^\tau)$. This intuitively suggests that when $\beta_1 = \beta_2$, Adam and Signum may behave similarly [3]. This motivates the conjecture that the main benefit of Adam over Signum is the fact that in Adam, $\beta_2$ can be varied independently of $\beta_1$. In Figure 1 we have $\beta_2 = 0.95$ and $\beta_1 = 0.9$ which are close, and as pointed out earlier, both optimizers have similar performance and stability.

We examine this hypothesis further in Figure 6 by varying $\beta_1$ and setting $\beta_2 = \beta_1$, and again find that Signum and Adam behave very similarly. However, we also note that when we vary $\beta_1$ for Adam while fixing $\beta_2$ we get more stability for $\beta_1$ as compared to Signum.

**Takeaway:** With $\beta_2 = \beta_1$ Adam and Signum behave similarly and the standard setting for training language models ($\beta_2 = 0.95, \beta_1 = 0.9$) is close to this.

## 3 INVESTIGATING THE IMPACT OF ADAPTIVITY FOR OPTIMIZER STABILITY AND PERFORMANCE

Ablations in the previous section revealed the striking similarity in performance and stability across multiple optimizers compared to Adam. Adam and its other variants are designed to have a high degree of adaptivity at a fine granularity (per-parameter learning rates) throughout the training process. This adaptivity is often credited with the stability and robust performance observed in these optimizers. However, a critical question arises: to what extent is this adaptivity needed for different parameters of the network? By identifying the necessity of adaptivity for different network components to ensure both performance and stability, we aim to discern whether simpler optimizers like SGD can achieve similar benefits with minimal modifications. Since higher momentum can often play the same role as a better preconditioner and to have all algorithms on an equal footing, we will fix $\beta_1 = 0.9$ for all optimizers in this section.

The main optimizer we study in this section is a "layer-wise" variant of Adam, which we coin as 'Adalayer'. We use Adalayer for our investigations because it lends a greater ease of understanding compared to full-fledged Adam in identifying parts of the network which may be particularly critical for optimizer performance and stability. Note that this layerwise variant is a special case of a previously known optimizer called Blockwise Adaptive Gradient with Momentum (BAGM) (Zheng & Kwok, 2019).

### 3.1 ADALAYER

To study the behavior of adaptive optimizers like Adam, we begin with describing a layer-wise version of Adam which we refer to as Adalayer. Adam, Adafactor and Adalayer all (approximately) store the diagonal second moment matrix, but with coarser and coarser granularity; for a layer of dimension $m \times n$, Adam explicitly maintains the second moment matrix using $mn$ parameters in the shape of a matrix. Adafactor stores row and column averages of the second moment matrix which serve as a rank-1 approximation to the second

---

**Algorithm 1:** Adalayer

**Parameters:** Learning rate $\eta$, exponential decay rates for the moment estimates $\beta_1, \beta_2$, number of steps $T, \epsilon$
**while** $t \leq T$ **do**
    **for** *each layer $l$ with $p$ parameters* **do**
        $g_t^l \leftarrow \nabla_l L(w_t)$ ;
        $v_t^l \leftarrow \beta_2 \cdot v_{t-1}^l + (1 - \beta_2) \cdot p^{-1/2} \cdot \|g_t^l\|_2^2$ ;
        $m_t^l \leftarrow \beta_1 \cdot m_{t-1}^l + (1 - \beta_1)g_t^l$ ;
        $w_{t+1}^l \leftarrow w_t^l - \eta \cdot \frac{m_t^l}{\sqrt{v_t^l} + \epsilon}$ ;
    **end**
**end**

---

[3]This is not true in theory, as the ratio can vary across parameters.

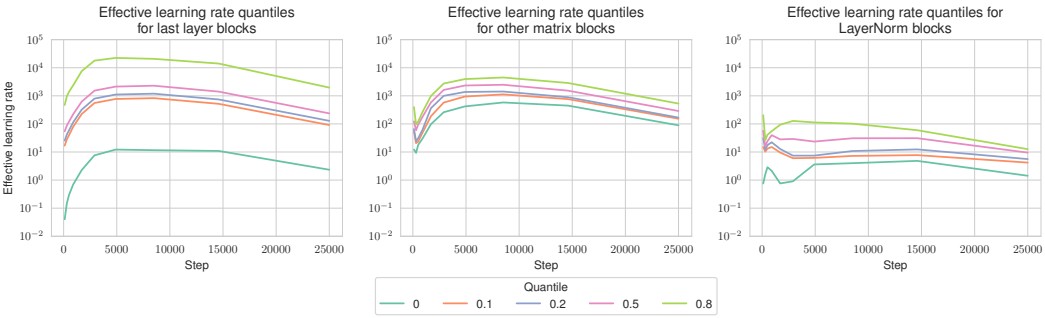

Figure 7: Quantiles of effective learning rates ($\eta_t/(\sqrt{v_t^l} + \epsilon)$ for each layer $l$) for the last layer blocks (**Left**), the LayerNorm blocks (**Right**), and the other matrix blocks (**Middle**) for a 150m model trained using Adalayer*. Unlike the other matrix blocks and LayerNorm parameters, the effective learning rates across logits vary across multiple orders of magnitude, providing evidence for the need to precondition them separately.

moment matrix. Finally, Adalayer stores a single scalar which is the average of the second moment matrix. We will later consider a generalization of Adalayer where instead of averaging second moment over a layer we will average it over a "block" of parameters which can be a subset of a layer. We note that similar algorithms have been studied before (Ginsburg et al., 2019; Agarwal et al., 2020) but we choose to study this variant since it is a direct analogue of Adam and Adafactor. A simplified version of Adalayer optimizer is given in Algorithm 1; other details such as bias correction are kept same as that for Adam. In Appendix C, we provide more details about our Adalayer implementation and a correction we make to how Adalayer treats the last layer in order to achieve similar performance and stability to Adam. Specifically, Adalayer when naively applied for each layer is neither performant nor stable to learning rate (Figure 10); however, if we additionally treat the set of weights in the last layer feeding into each logit as its own block, this recovers most of the performance and stability of Adam (see the dotted blue lines in Figure 8). We henceforth refer to Adalayer with this correction as **Adalayer***.

To study how Adalayer* preconditions the network, we plot effective learning rates used for different logits by Adalayer* in Figure 7 (**Left**). Here, the effective learning rate for a layer $l$ in the network is $\eta_t/(\sqrt{v_t^l} + \epsilon)$. We find that Adalayer* indeed uses vastly different learning rates for different logits, supporting our hypothesis that preconditioning weight in different logits separately is important for performance and stability.

### 3.2 BOTH THE LAST LAYER AND LAYERNORM PARAMETERS NEED ADAPTIVITY

The results using Adalayer* in the previous section suggest that all layers except the last layer only need a iteration-dependent scalar correction to their learning rate. We now ask a stronger question: do we need these scales at all? Or can we train the remaining layers with SGD? This hypothesis is supported by looking at Figure 7 (middle) where we observe that the learning rates for different matrix layers (except the last layer) assigned by Adalayer* are remarkably similar.

To test this, we train the last layer with Adalayer* (fixing a learning rate of $3.16e - 3$) and the rest of the layers with SGD, both with $\beta_1 = 0.9$. In Figure 8 (**Left**) we show the results while sweeping over SGD learning rates from $0.1$ to $3160$. While this improves upon the performance of SGD, we do not recover stability of the Adalayer* and Adam baselines. We trace this instability to *LayerNorm blocks*: Figure 7 (**Left**) shows that the effective learning rates for the LayerNorm blocks are much smaller, which suggests that they may destabilize at higher SGD learning rates. To ameliorate this, in Figure 8 (**Right**) we add LayerNorm parameters to those being trained with Adalayer* and find that this is sufficient to recover both performance and stability of Adalayer*. In Figure 11 in the Appendix, we see this trend continue to hold for 300m and 600m parameter models.

We conduct additional experiments investigating these 'hybrid' variants of SGD in the Appendix. Despite most language model parameters being in matrix layers, we show that if we only apply Adalayer* on the matrix layers and train the last layer and LayerNorm layers with SGD, this does not

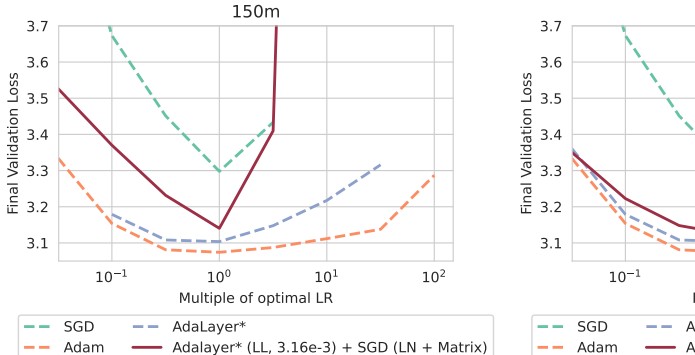

Figure 8: (**Left**): Training the last layer using Adalayer* with a fixed learning rate of $3.16e - 3$ and other LayerNorm and matrix blocks using SGD achieves better performance than SGD, but does not recover stability. (**Right**): Training both the last layer and LayerNorm blocks using Adalayer* and the other matrix blocks using SGD nearly recovers or exceeds performance of Adalayer*, and achieves stability across learning rates. Dotted lines are baselines from optimizers previously given in Sections 2 and 3.1.

recover performance and stability. Another plausible remedy to address the small effective learning rates of the LayerNorm blocks is to simply not train these LayerNorm parameters. In Appendix D, Figure 13, we show that this improves performance and stability relative to SGD but does not recover Adalayer* performance. Finally in Appendix E, we replace SGD with AdaSGD (Wang & Wiens, 2020) which still uses a global learning rate but is now adaptive through training; in Figure 16 we show AdaSGD + Adalayer* can recover full Adalayer* performance, but a small gap remains to recover Adam performance.

We acknowledge that while adaptivity for the last layer and LayerNorm parameters seems *necessary* to retain optimizer performance and stability, there remains a small gap which is fulfilled by adaptivity on the other network parameters; in Figure 14 we also try training 150m and 300m models using SGD on the matrix blocks and *Adafactor* on the last layer and LayerNorm blocks; we find that performance and stability is comparable, but does not exceed that of Adafactor on the entire network.

Further, note that a caveat of the above results is that we have introduced an additional hyperparameter— SGD learning rate, which we are sweeping over— while keeping the Adalayer* learning rate in the last layer and layer norm layers fixed. While decoupling the learning rates here is needed (due to SGD's performant learning rates being orders of magnitude higher than that of Adalayer*), this may be responsible for the observed stability. To address this, we perform the following experiment: we train all the layers with Adalayer* (sweeping over learning rate $\eta$) but we *stop updating the second moment estimates* for all layers *except the last layer and LayerNorm blocks* after initialization. This implies that these layers are effectively being trained by SGD with a fixed learning rate, though unlike the above results, these learning rates are different for different layers. We implement this by passing 1000 batches to initialized 150m and 300m models to obtain second moment estimates for all layers without letting the model take a gradient step, and then allowing the model to train as normal under the same settings as all of our ablations. As in our previous investigation, we fix other hyperparameters to be the same: $\beta_1 = 0.9$, $\beta_2 = 0.95$ and $\epsilon = 1e - 15$.

In Figure 9 we show the resulting learning rate sweep for freezing Adalayer* learning rate scales at initialization (with the exception of the last layer and LayerNorm). Surprisingly, we find for the 150m model that we can almost entirely recover the stability and performance of Adalayer*. For the 300m model we also match or exceed the performance of Adalayer*, and even nearly match the peak performance of Adam. Note again that this sweeps learning rate across all network parameters. This provides further evidence for the importance of adaptivity in the last layer and LayerNorm, where in contrast we could have used fixed ratios from initialization for all other parameters to recover the performance and stability of Adalayer*. In Appendix F we present additional experiments which include allowing Adalayer* to update for only one of the last layer or LayerNorm, or turning off LayerNorm training. None of these partial modifications fully recover performance and stability.

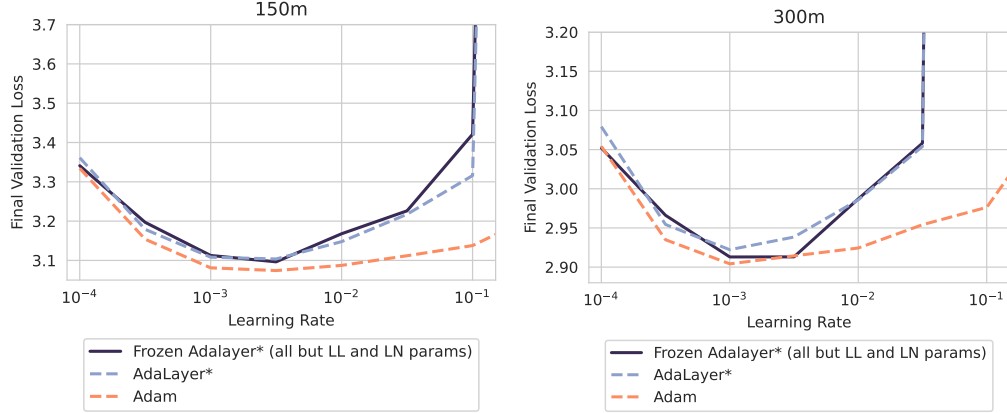

Figure 9: Training 150m (**Left**) and 300m (**Right**) models using fixed Adalayer* learning rate ratios from initialization, with the exception of last layer and LayerNorm parameters. This almost entirely matches the performance and stability of Adalayer* in the 150m model, and exceeds Adalayer*'s peak performance to be comparable with Adam.

## 4 DISCUSSION AND LIMITATIONS

After a comprehensive comparison of a variety of optimizers for language modeling, we have found that many optimizers seem to be roughly equivalent both in terms of optimal performance and hyperparameter stability. The only differing axis for practitioners to consider is thus memory constraints, but based on our results, memory considerations for optimizer choice should not affect the performance of their training run, at least across the optimizers tested (i.e. optimizers using diagonal preconditioning). For tuning guidelines, we have reported the optimal learning rates in Appendix B found in our sweeps as a starting point for practitioners in tuning. We also observed that most other parameters are very stable for the optimal learning rate (with exception to momentum) and thus we believe tuning learning rate and momentum should be prioritized if one has a limited compute budget for hyperparameter sweeping.

Diving deeper, we have shown that the treatment of the last layer and LayerNorm parameters is crucial for realizing the benefits of adaptive optimizers. This highlights the following research directions as the most promising for developing optimizers for language model training. Firstly, our results suggest that within the realm of diagonal preconditioning optimizers, there may be room for more memory-efficient alternatives to AdamW to be designed beyond using factorization techniques (eg. Adafactor with momentum and Lion from our initial sweeps). We have provided one direction for cutting down on memory overhead, where we have shown that the need for adaptivity is distributed unevenly across different parameters of the network (with similar ideas already explored in concurrent work (Zhang et al., 2024c)). Secondly, in terms of improving the performance of these models, our work suggests that optimizer choice is unlikely to be a promising approach— if we are limited to diagonal preconditioning optimizers. Thus, it is likely that we need to turn to non-diagonal methods to improve on our current methods; this is indeed where previous work has demonstrated improved performance to Adam (Gupta et al., 2018a).

Of course, there are several limitations to our study including the fact that due to computational constraints we only ablate a few architecture decisions, that we only consider one dimensional hyperparameter sweeps, we fix batch size, and that we limit our study to autoregressive language modeling with a single dataset. Despite these limitations, we believe that the study sheds new light on the fundamentals of optimization for language modeling.

## ACKNOWLEDGMENTS

SK, DM and RZ acknowledges support from the Office of Naval Research under award N00014-22-1-2377 and the National Science Foundation Grant under award #IIS 2229881. This work has been made possible in part by a gift from the Chan Zuckerberg Initiative Foundation to establish the Kempner Institute for the Study of Natural and Artificial Intelligence. NV, DM and RZ are supported by a Simons Investigator Fellowship, NSF grant DMS-2134157, DARPA grant W911NF2010021,and DOE grant DE-SC0022199. RZ and DM are supported by Kempner Institute Graduate Research Fellowships.

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

# A   RELATED WORK

One closely related work to ours is Wortsman et al. (2024), which explores the stability of Adam with respect to learning rate. We extend the comparison to other optimizers including SGD, Lion and Adafactor, as well as other hyperparameters including momentum and weight decay.

**Optimizers:** SGD (Robbins & Monro, 1951) had been the workhorse optimizer for deep learning until 2015, when Adam (Kingma & Ba, 2015) was introduced. Adam is a diagonal preconditioning algorithm that maintains a per-parameter learning rate. Over time, coarser variants of Adam have been proposed, which do not explicitly maintain a learning rate per parameter. Adafactor (Shazeer & Stern, 2018; Zhai et al., 2022) maintains a rank-1 approximation of the preconditioner matrix of Adam. Previous works have also explored Signum (Bernstein et al., 2018; 2019) and have observed its benefits in terms of communication efficiency and fault tolerance. Other works have also explored the similarity of Adam with variants of Signum (Balles & Hennig, 2018), and recently, a close variant of Signum, called Lion (Chen et al., 2023), was discovered using symbolic search over algorithms. Some other optimizers that have recently gained increasing attention from the community include Shampoo (Gupta et al., 2018b) and Sophia (Liu et al., 2024).

Similar to us, prior work has explicitly studied optimizers comparisons across different domains and architectures. For instance, Schmidt et al. (2021) performs a sweep over optimizers, schedulers, and seeds over various vision tasks. As mentioned in the introduction, Kaddour et al. (2024) compares optimizers for masked language modeling and at a fixed model scale. Other works have called to attention better benchmarking practices for optimizers (Schneider et al., 2019; Sivaprasad et al., 2020; Bartz-Beielstein et al., 2020).

**Adam and Signum:** Many works have explored the relationship between Adam and variants of Signum (Balles & Hennig, 2018; Balles et al., 2020; Kunstner et al., 2023) and empirically demonstrated that Signum (or its close variants) generally performs comparably to Adam. Balles et al. (2020) also argued that signSGD generally performs better when the Hessian is close to diagonal, however, it is unclear if this holds for practical settings. Kunstner et al. (2023) recently demonstrated that Adam and a close variant of Signum exhibit similar performance on a variety of datasets including WikiText-2 (Merity et al., 2017) and SQuAD (Rajpurkar et al., 2016). However, in contrast with our work, all of these are restricted to the setting of vision or masked language modeling, and generally do not sweep over multiple hyperparameters.

**Layerwise or blockwise Adam:** We study Adalayer, a layerwise version of Adam. This is a special case of the BAGM optimizer (Zheng & Kwok, 2019), specifically BAGM B.1. Similar algorithms have also been studied by previous works (Ginsburg et al., 2019; Agarwal et al., 2020; Liu et al., 2021; Zhang et al., 2024c). In particular, concurrent to our work, Zhang et al. (2024c) propose an algorithm termed Adam-mini, which closely tracks a modified version of Adalayer (called Adalayer*), and demonstrate comparable performance to AdamW. Note that, in our work, Adalayer* is introduced to understand the role played by preconditioning in Adam, and we do not specifically focus on the final performance. Zhang et al. (2024b) empirically study the Hessian spectrum of transformers at initialization and find it to be more heterogeneous across layers as compared to ResNets. They argue that this heterogeneity is evidence towards the importance of Adam in training transformers. In contrast our results (Section 3.2) show that Adam's preconditioning is particularly important for the last layer and LayerNorm parameters to achieve performance and learning rate stability.

**Other related works:** For vision transformers, in the fine-tuning phase, Kumar et al. (2024) show that using SGD with frozen embedding parameters leads to competitive performance with Adam. Jelassi et al. (2022) explore the similarity between Adam and normalized gradient descent (Nesterov, 2004) and show that normalized gradient descent on GANs does not suffer from mode collapse, while SGD does. Jiang et al. (2023) empirically demonstrate that Adam steers the parameter trajectory towards better-conditioned regions than SGD. Pan & Li (2022) also show that the parameter trajectory of Adam exhibits much higher directional smoothness than that of SGD. Ahn et al. (2024) show that the performance gap between Adam and SGD exacerbates with depth of the network. In a similar vein to us, Kunstner et al. (2024) show that Adam is less sensitive than gradient descent to class-imbalance present in language tasks; we provide further evidence for the importance of preconditioning the last layer, as well as the LayerNorm parameters.

# B ADDITIONAL MAIN SWEEP RESULTS

| Optimizer | Optimal Learning Rate |
|---|---|
| Adam | 3.16e-3 (150m), 1e-3 (300m), 1e-3 (600m), 1e-3 (1.2b) |
| Adafactor | 3.16e-3 (150m), 1e-3 (300m), 1e-3 (600m), 1e-3 (1.2b) |
| Lion | 3.16e-4 (150m), 3.16e-4 (300m), 3.16e-4 (600m), 1e-4 (1.2b) |
| Signum | 3.16e-4 (150m), 3.16e-4 (300m), 3.16e-4 (600m), 3.16e-4 (1.2b) |

Table 1: Optimal learning rates for various optimizers from Figure 1.

For our ablations in Figure 1, we report on the optimal learning rate found for each optimizer in Table 1. In general, we find that the optimal learning rate for Adam and Adafactor are similar, with the optimal learning rate of Lion and Signum an order of magnitude smaller.

# C ADALAYER

As mentioned in Section 3, to investigate the role of preconditioning on language models for optimizers like Adam, we introduce the Adalayer optimizer for ease of analysis. In this section, we first establish the performance and stability of Adalayer as a reasonable proxy for Adam by making a modification to how Adalayer treats the last layer of the network.

In Figure 10 we study the behavior of Adalayer across learning rates. To preserve the correspondence with Adam we fix other hyperparameters to be the same: $\beta_1 = 0.9$, $\beta_2 = 0.95$ and $\epsilon = 1e - 15$. We find that Adalayer has better performance than SGD, but it performs worse than Adam and also lacks Adam's stability across learning rates. The major difference between Adam and Adalayer is the preconditioning done by Adam *within* a layer. Intuitively, this preconditioning will have large effects in layers where we expect different weights within a layer to have different gradient scales. The first candidate for such a layer is the *last layer*, since different tokens have widely different frequencies leading to different gradient scales. To test this hypothesis, we run a corrected version[4] of Adalayer where we treat the set of weights feeding into a logit as a separate block. We henceforth refer to Adalayer with this correction as **Adalayer\***. This is plotted in Figure 10 and we observe that Adalayer\* almost recovers the performance as well as a large fraction of the stability of Adam.

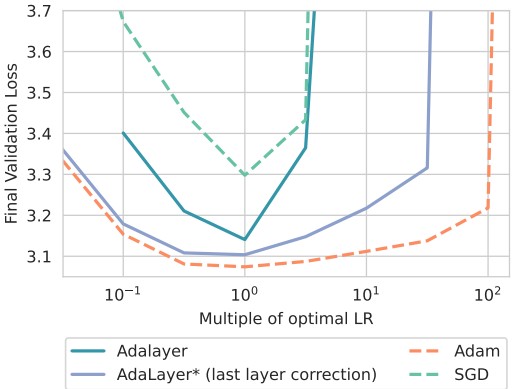

Figure 10: Modifying Adalayer with the last layer correction improves performance and stability across learning rates.

---

[4]We note that this reasoning also applies to the first layer, but in our ablations applying this correction the first layer did not make a significant difference.

# D  ADDITIONAL EXPERIMENTS: SGD + ADAPTIVE VARIANTS (ADALAYER*, ADAFACTOR)

In this section, we report additional experiments involving training language models with SGD on a fraction of the models' parameters and an adaptive optimizer on the remaining parameters. Firstly, we show that our results from Section 3.2 hold even when training 600m parameter models with Adalayer* applied only on the last layer and LayerNorm parameters in Figure 11.

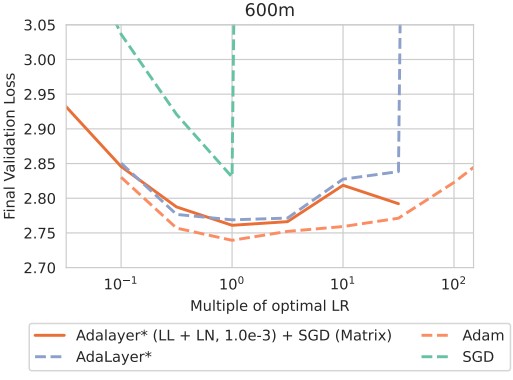

Figure 11: As in Section 3.2, we train 300m (**Left**) and 600m models (**Right**) with Adalayer* on the last layer and LayerNorm parameters, and train the remaining model parameters with SGD. We see that performance and stability continues to match that of Adalayer* or outperform Adalayer* even at these larger scales..

We also provide further ablations supporting our claim that the largest impact of the adaptivity of Adalayer* is concentrated on the last layer and LayerNorm parameters. Firstly, we train 150m models using Adalayer* on only the matrix parameters, while training the last layer and LayerNorm parameters with SGD. In Figure 12, we see performance improves relative to SGD but we see similar instability at larger learning rates.

Secondly, given that the effective learning rates of the LayerNorm blocks were observed to be small in Figure 7 (**Right**), it is reasonable to ask whether training the LayerNorm parameters is necessary at all; in Figure 13, we show results for training 150m and 300m models using Adalayer* only on the last layer, using SGD on all other matrix blocks, and turning off training for the LayerNorm parameters. This indeed yields greater stability in comparison to Figure 8 (**Left**) but does not fully recover the performance of Adalayer*, indicating that training LayerNorm parameters helps with performance, which seems more pronounced in the larger model.

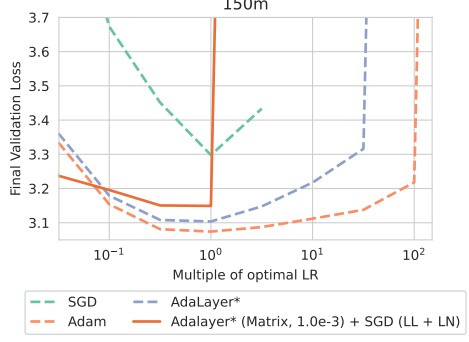

Figure 12: We train 150m models using Adalayer* on the matrix layers with a fixed learning rate of $1e-3$ and using SGD on the last layer and LayerNorm parameters. Compared to the results in Figure 8, we do not recover the same stability nor do we reach the optimal performance of Adalayer*.

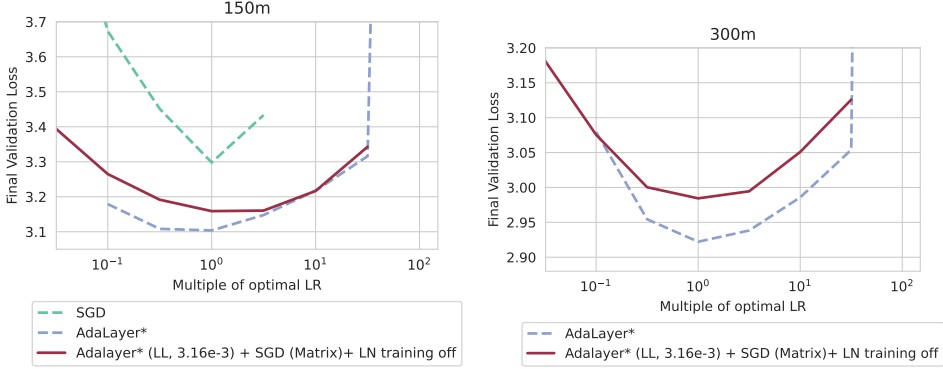

Figure 13: Training 150m (**Left**) and 300m (**Right**) models using Adalayer* on the last layer with a fixed learning rate of $3.16e-3$ and using SGD on other matrix blocks, while turning off the option to train LayerNorm parameters. We see that while the performance and stability is improved compared to SGD, it is still not as performant as Adalayer*. This indicates a degree of importance of training LayerNorm parameters for these models.

We saw in Figure 8 (**Middle, Right**) that using Adalayer* on only the last layer and LayerNorm parameters sufficed to recover or exceed the performance of Adalayer*. In Figure 14, we report a learning rate sweep over the analogous experiment but using Adafactor on the last layer and LayerNorm parameters with a fixed learning rate. For the 150m model, using a learning rate of $3.16e-3$ with Adafactor yielded better performance than Adafactor for low learning rates, and is comparable in terms of performance and stability. For the 300m model, the difference between Adafactor and our 'hybrid' optimizer is more distinct at higher learning rates for fixed Adafactor learning rate $1.0e-3$ and $3.16e-3$, but is comparable until the peak validation loss.

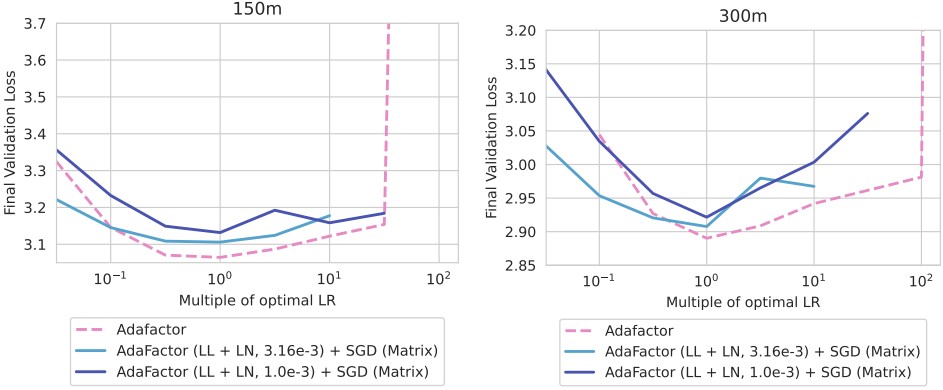

Figure 14: Training 150m (**Left**) and 300m (**Right**) models using Adafactor on the last layer with a fixed learning rate and using SGD on other matrix blocks. We see that performance and stability is comparable to Adafactor, but does not exceed it, particularly at higher learning rates.

## E   ADDITIONAL EXPERIMENTS: ADASGD + ADALAYER*

The previous experiments on SGD + Adalayer* show that although using a fixed learning rate for the matrix parameters largely recovers performance of full Adalayer*, there remains a small gap to full Adalayer* or Adam performance. Can we recover what is remaining? We further investigate the importance of adaptivity at different levels of granularity by training the matrix parameters with AdaSGD (Wang & Wiens, 2020) instead of SGD. AdaSGD still uses a global learning rate but allows this scalar to be adaptive across training— in other words, this is a "global" version of Adalayer* where all matrix parameters are treated as one block.

In Figure 15 we conduct an analogous learning rate sweep for AdaSGD for $\beta_0 \in \{0.9, 0.98\}$ as we did in Section 2. The performance of AdaSGD is marginally better than SGD, but still suffers from learning rate instability; this is consistent with our previous findings in Figure 7, where we saw that certain parameters had effective learning rates which were multiple orders of magnitude different from other parameters, and thus a single adaptive learning rate across all parameters likely would not suffice for achieving performance and stability.

In Figure 16 we replace SGD with AdaSGD on the matrix parameters and use Adalayer* to train the last layer and LayerNorm parameters. The performance now essentially matches that of full Adalayer* compared to SGD + Adalayer*, showing that the single adaptive learning rate on matrix parameters is largely sufficient to recover Adalayer* performance. However, a gap still remains between Adam performance even with an adaptive learning rate across matrix parameters; it is likely that this small gap is attributed to the need for increased adaptivity granularity for certain matrix parameters.

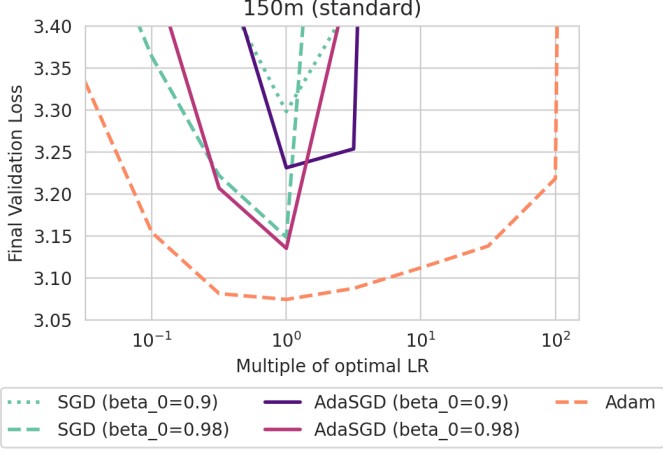

Figure 15: Learning rate sweep for AdaSGD with two $\beta_0$ values. We observe that AdaSGD performs marginally better than SGD but still lacks learning rate stability.

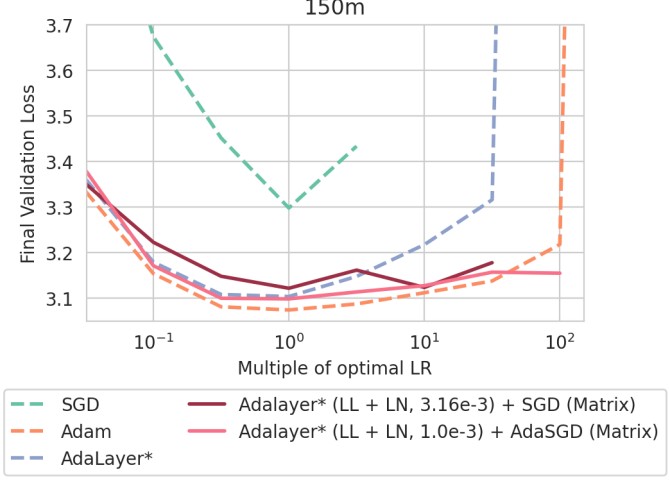

Figure 16: When replacing SGD with AdaSGD on the matrix parameters and using Adalayer* to train the last layer and LayerNorm parameters, the performance and stability has improved to match that of Adalayer*, but a gap still remains between Adam performance even with an adaptive learning rate across matrix parameters.

## F ADDITIONAL EXPERIMENTS: FREEZING ADALAYER LEARNING RATE RATIOS

In this section, we report additional experiments exploring whether the results involving frozen Adalayer* in Section 3.2 need both last layer and LayerNorm adaptivity. We show that this is indeed the case by conducting the same sweep for frozen Adalayer* while trying to also freeze the learning rate ratios for last layer or LayerNorm parameters as well. In Figure 17, we show that fixing initialized learning rate ratios for *all* layers does not reach peak performance of Adalayer*, nor does it exhibit stability. In Figure 18, we show that either continuing to update the LayerNorm parameters or the last layer parameters can achieve the peak performance of Adalayer* but is still unstable. Finally, we show results for turning off LayerNorm training while fixing learning rate ratios (with the exception of the last layer) in Figure 19. We conclude that it is necessary to maintain adaptivity for *both* the last layer and LayerNorm parameters, but understanding why the fixed ratios do not suffice would be an interesting question for future work.

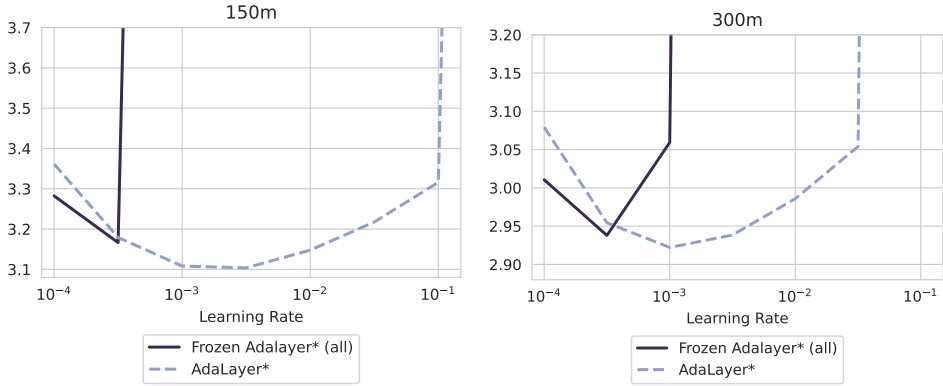

Figure 17: Training 150m (**Left**) and 300m (**Right**) models using fixed Adalayer* learning rate ratios from initialization for *all layers*. We observe this quickly diverges, achieving neither peak performance nor stability.

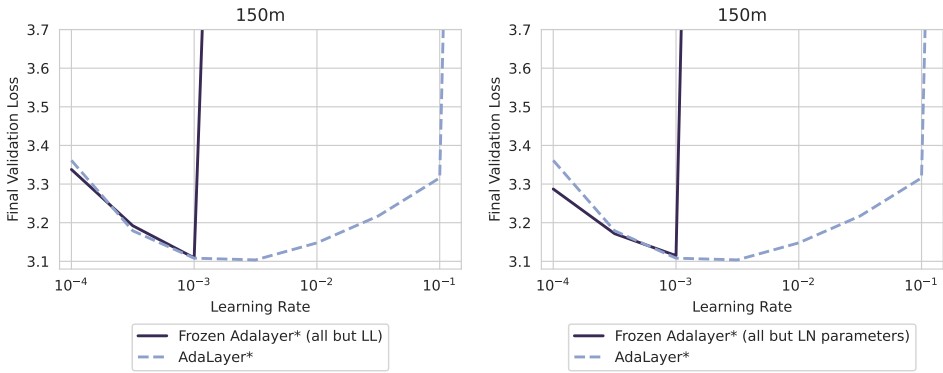

Figure 18: Training 150m models using fixed Adalayer* learning rate ratios from initialization while either excluding only the last layer (**Left**) or excluding only the LayerNorm parameters (**Right**). We observe both modifications reach peak performance but fails to be stable at higher learning rates.

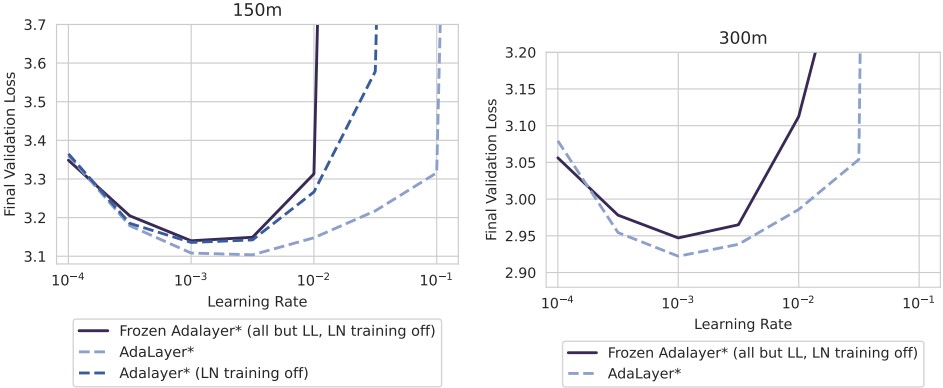

Figure 19: Training 150m (**Left**) and 300m (**Right**) models using fixed Adalayer* learning rate ratios from initialization, while letting the last layer continue to update, and turning LayerNorm training off. Stability across learning rates has improved but is less performant; for the 150m model we also plot the sweep for regular Adalayer* with LayerNorm training off, and we see that it is worse in performance compared to Adalayer* with LayerNorm training.

## G  SOPHIA

In this section, we compare Sophia (Liu et al., 2024) to Signum. Note that Signum is a special case of Sophia, achieved by setting $\rho = 0$. We find that Sophia does not outperform Signum. No significant change in performance was observed when transferring the hyperparameters suggested by Liu et al. (2024) (eg. $\beta_1$, $\beta_2$, $\varepsilon$, weight decay), nor when additionally scaling attention by the inverse of layer index which was used in the original Sophia implementation.

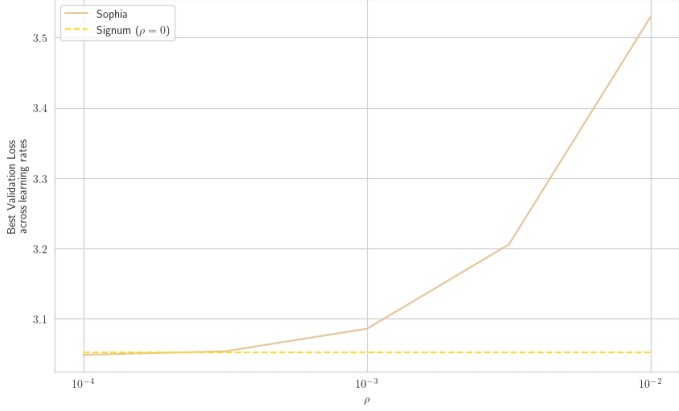

Figure 20: Comparing Sophia (Liu et al., 2024) and Signum for the 150M model in our default setup.

