# OpenReview forum: "Deconstructing What Makes a Good Optimizer for Autoregressive Language Models"
_ICLR.cc/2025/Conference — ICLR 2025 Poster_

### Official Review · Reviewer_4oK2 · 2024-11-03

**Soundness:** 3
**Presentation:** 3
**Contribution:** 2
**Rating:** 5
**Confidence:** 3

**Summary:**

This paper evaluates several optimizers, including SGD, Adam, Adafactor, Lion, and Sophia, in the context of large language models (LLMs). The results suggest that all optimizers, except for SGD, perform similarly regarding optimal performance and sensitivity to hyperparameters. The authors then further investigated two simplified versions of Adam: Signed Momentum and AdaLayer. They found that adaptivity in the LayerNorm and final layers is essential for achieving stable performance and learning rates.

**Strengths:**

- The evaluation of hyperparameters is thorough, and the documentation of experimental details is complete. Although it is limited to one-dimensional changes and does not capture the interplay between hyperparameters, the authors clearly state these limitations.

- The observation that adaptivity in the LayerNorm and final layers is necessary for LLMs is interesting.

**Weaknesses:**

- My main concern with this paper is the significance of its contribution. The paper does not introduce new algorithms or provide insights into why some evaluated algorithms work or fail; it is limited to an empirical comparison. While this can still be valuable, I’m unsure if the paper adequately addresses the question posed in its title, “What Makes a Good Optimizer for Autoregressive Language Models.” It also does not seem to explain how the characteristics of “autoregressive” or “language” models interact with optimization or why the results might differ in other tasks and architectures. Additionally, the organization is somewhat confusing, and it’s unclear how the two parts of the paper relate.

- There are quite a few observations from each experiment, but it’s unclear what the main message or takeaway of the paper is. For example, it doesn’t clearly outline what practitioners should do, what future researchers in algorithm design should focus on, or provide any conceptual or theoretical insights that explain these observations. The paper could be significantly improved by clarifying the main questions it actually addresses and having a better discussion paragraph on what others could do with this information.

- Citations of other optimizer-comparison papers could be more comprehensive. For instance, [Schmidt et al., 2020: *Descending through a Crowded Valley - Benchmarking Deep Learning Optimizers*] is an example that could be included.

**Questions:**

- Many follow-up works aim to improve Adam's performance or to develop memory- or compute-efficient versions of it. What is the rationale behind selecting these specific optimizers for comparison? Was the choice based on practical popularity, conceptual connections to Adam, or some other criteria?

- The plots comparing final validation loss (e.g., Figure 1) are presented so that each optimizer’s optimal learning rate aligns, with the x-axis showing multiples of this optimal learning rate. However, why should different optimizers be compared over the same scale of learning rate values? For example, SGD appears the most sensitive to changes in learning rate, but could this be due to it requiring a finer-grained learning rate grid for a fair comparison?

- A concurrent work, [Zhang et al. 2024: Adam-mini: Use Fewer Learning Rates To Gain More], also explores the concept of a coarser, "parameter-group" -wise learning rate for Adam, proposing it as the minimal level of adaptivity required. While it is of course not required to cite or evaluate concurrent work, a comparison with AdaLayer would be interesting given the similarity in approach. It would also be useful to see whether *Adam-mini* aligns with this paper’s findings on layer-specific adaptivity. This paper shows that adaptivity in the last layer and LayerNorm is necessary, yet limiting adaptivity to these layers alone still underperforms compared to Adam. So, what degree of adaptivity is "sufficient" to achieve Adam's full performance?

- Minor changes are needed on some of the later plots, as they are difficult to read due to small font sizes in the legends and axes, and they lose resolution when zoomed in. Please adjust these for clarity.

---

> ### Author Response · Authors · 2024-11-19
> **Response to Reviewer (1/2)**
>
> We thank the reviewer for their review and for recognizing our work’s findings as interesting. We’d like to address the issues brought up by the reviewer, as well as answer their specific questions.
>
> ### Weaknesses:
> 1. **Limited Contribution**: We acknowledge that our paper is an empirical comparison, and we appreciate your recognition of its value as a resource providing a comprehensive comparison across optimizers in a standardized codebase. While our scope is indeed limited to autoregressive language models, we believe the paper offers key takeaways for practitioners, such as the importance of adaptivity in the last layer and layer norm parameters for maintaining performance and stability, particularly relative to Adam. If the title appears misleading, we are very open to suggestions on how to make it more specific and reflective of these findings. Regarding the relationship between the two parts of the paper, our goal was to bridge the observed performance gap between SGD and other diagonal optimizers identified in the first section. We approached this by empirically examining the parameters requiring adaptivity, offering insights into why algorithms like SGD fail—namely, its inability to assign appropriate learning rate scales to these critical parameters. We will work to clarify the connection between these two sections in the revised version.
> 2. **Main takeaways from our work**: Thank you for bringing this up; indeed, one aim of our work is to provide a source for practitioners to refer to when comparing across various standard optimizers and hyperparameters. We’d like to provide an outline below addressing each point you have mentioned (eg. best practices for practitioners, conceptual/theoretical insights, and most fruitful further directions in algorithm design). We agree that the main takeaways from our results could have been consolidated and communicated more clearly; we will add a paragraph in the Discussion section of the revised pdf including these points below.
>
> The primary takeaway from our ablations is that across the optimizers we tested, there is no ‘better optimizer’ based on both optimal performance and hyperparameter stability. The only differing axis for practitioners to consider is thus memory constraints, but based on our results, they can be reassured that memory considerations for optimizer choice should not affect the performance of their training run, at least across the optimizers tested (i.e. optimizers using diagonal preconditioning). For more concrete takeaways regarding our hyperparameter sweeps, we reported the optimal learning rates found in our sweeps as a starting point for practitioners in tuning (eg. tuning optimizers like Signum and Lion seem to require learning rates an order of magnitude smaller compared to Adam/Adafactor). In terms of other tuning practices, we observed most other parameters are very stable for the optimal learning rate (with exception to momentum) and thus we believe tuning learning rate and momentum should be prioritized if one has a limited compute budget for hyperparameter sweeping.
>
> Our work highlights the following research directions as the most promising for developing optimizers for language model training. Firstly, our results suggest that within the realm of diagonal preconditioning optimizers, there may be room for more memory-efficient alternatives to AdamW to be designed (and has already been explored in concurrent work [1]) for domains where one is compute-constrained; we have provided one direction for cutting down on memory, where we have shown that the need for adaptivity is distributed unevenly across different parameters of the network. Secondly, in terms of improving the performance of these models, our work suggests that optimizer choice is unlikely to be a promising approach– if we are limited to diagonal preconditioning optimizers. Thus, it is likely that we need to turn to non-diagonal methods to improve on our current methods; this is indeed where previous work has demonstrated improved performance to Adam [2].
>
> 3. **Related work on optimization comparisons**: Thank you for mentioning this, we have revised the manuscript to include prior literature conducting empirical optimizer comparisons across various settings.
>
> [1] Zhang, Yushun, et al. "Adam-mini: Use fewer learning rates to gain more." (2024).
>
> [2] Gupta, Vineet, Tomer Koren, and Yoram Singer. "Shampoo: Preconditioned stochastic tensor optimization." (2018).

---

> > ### Author Response · Authors · 2024-11-19
> > **Response to Reviewer (2/2)**
> >
> > ### Questions:
> > 1. **Regarding our choice of optimizers for comparing to Adam**: The optimizers that we chose to study in this work are those which also use diagonal preconditioning and have seen common adoption in practice (eg. Lion, Adafactor, Signum); thus, as you have already mentioned, we primarily selected these based on popularity and a conceptual connection to Adam. For the later sections of our work, we were not aiming to design a new optimizer that exceeds Adam performance, and instead wanted to better understand the effect of adaptivity across different parameters of the network; as a first step in this direction, we chose to group parameters by layer types, which identifies Adalayer* as a natural choice that is an approximation of Adam while being conceptually similar.
> > 2. **Regarding fixing the scale of the range of learning rate values for different optimizers**: We believe having the same scale of the learning rate grid across optimizers is the fairest comparison for evaluating hyperparameter stability. Our comparisons are meant to show that for large scale runs— where optimal hyperparameters are generally selected by fitting scaling laws— it is very likely that SGD's performance is much more suboptimal as compared to others (for a similar error in the estimation of optimal learning rate).
> > 3. **Regarding Adam-mini**: Indeed, Adam-mini closely tracks Adalayer* in our work and provides concurrent evidence that one can reduce the number of learning rates that Adam provides and yield similar performance with significant memory improvements. We’d like to note that in our work we find that Adafactor (with momentum) and Lion were comparable in performance stability to Adam, which are already more compute-efficient than Adam-mini. To answer your question of “what degree of adaptivity is "sufficient" to achieve Adam's full performance” along the axis of Adalayer* and Adam-mini (which tries to identify a subset of parameters requiring adaptivity at some granularity), you are correct that the remaining performance gap could be obtained from adding adaptivity for some matrix parameters; from the Hessian structure perspective, Adam-mini suggests certain candidates, such as having a learning rate assigned to each attention head’s query and key matrix, or a learning rate assigned to MLP parameters grouped via output tokens. To our knowledge, Adam-mini doesn’t ablate the individual absence or addition of adaptivity on such parameters, but this would be an interesting direction.
> > 4. **Regarding the plots**: Thank you for pointing this out, we will enlarge the font sizes in the legends and axes of our plots for improved clarity in the revision.
> >
> > We hope that we adequately addressed your concerns and questions, and are happy to provide any additional clarifications or engage in further discussion.

---

> > > ### Author Response · Authors · 2024-11-24
> > >
> > > We would like to thank the reviewer again for their valuable feedback and suggestions. We would be grateful if the reviewer would consider updating their score based on our response. If not, we would be happy to answer any additional questions.

---

> > > > ### Author Response · Authors · 2024-12-01
> > > >
> > > > As the discussion period ends tomorrow, we would like to thank the reviewer for their feedback and kindly ask them to let us know if our responses have addressed their questions and concerns. If so, we would be grateful if the reviewer could update their score accordingly.

---

### Official Review · Reviewer_iK1k · 2024-11-04

**Soundness:** 2
**Presentation:** 3
**Contribution:** 1
**Rating:** 5
**Confidence:** 2

**Summary:**

The authors argue that the Adam optimizer is the default choice for most practitioners training language models, and they seek to evaluate this belief by comparing Adam against other popular optimizers. Their findings show that, except for SGD, other optimizers perform on par with Adam. Following this, they explore what components contribute to Adam's success by evaluating two simplified versions of it: Signed Momentum and Adalayer.

**Strengths:**

The Authors show that most optimizers, other than SGD, can match Adam in stability and performance across varied model sizes and hyperparameter settings. Furthermore, it reveals that, to recover stability concerning learning rate and to maintain strong performance, adaptivity is essential, particularly in the last layer and LayerNorm parameters. This insight challenges the notion that Adam is uniquely suited for language model training.

**Weaknesses:**

- Limited contribution: The main contribution of the paper is the finding that adaptivity on both the last layer and LayerNorm provides the necessary conditions for retaining performance and stability with respect to the learning rate.

- The paper could serve as a guide for those looking to choose an optimizer by comparing performance versus computational cost. However, after reading it, I didn’t feel I gained a clear insight to help decide on a better optimizer. In this sense, the paper has potential but feels incomplete.

**Questions:**

- line 165, are there only two hyperparameters in the problem? Because if there are more, then the optimal solution is only achievable by solving the N-D problem, with N being the number of hyperparameters.

- While I agree with the authors that cross evaluating all combinations of hyperparameters is intractable, why not use Bayesian optimization to find the best set of hyperparameters for each experiment?

---

> ### Author Response · Authors · 2024-11-19
> **Response to Reviewer (1/2)**
>
> We thank the reviewer for their review and recognizing that our results challenge the prevailing notion of Adam being uniquely suited for language model training. We’d like to address the issues brought up by the reviewer, as well as answer their specific questions.
>
> ### Weaknesses:
> 1. **Limited contribution**: We would like to reiterate the main contributions of our work and highlight their broader implications. Our paper provides a comprehensive empirical evaluation of optimizers for autoregressive language model pretraining, including a wide range of hyperparameters and model scales. One key finding is that no single algorithm consistently outperforms others in terms of both performance and hyperparameter stability, challenging the prevailing notion of Adam as the default optimizer for language models. In previous work, as we have noted in the manuscript, empirical studies of optimizers across hyperparameters have been either outside of the autoregressive language modeling domain, limited to Adam only, or limited to a fixed scale (see, eg. [1,2,3]). Additionally, as you noted, we show that adaptivity in both the last layer and LayerNorm parameters is essential for maintaining performance and learning rate stability. This insight not only informs optimization design but also suggests pathways for developing more memory-efficient optimizers—a direction that concurrent work has already started to explore. We believe our findings can encourage groups with more computational resources or those working in other domains to replicate and extend our results, scaling to larger models and exploring new designs. We would appreciate further clarification on why the reviewer views our contribution as limited, and are happy to discuss this point further.
> 2. **Gaining more insight about choosing optimizers**: Indeed, one aim of our work is to provide a guide for practitioners to refer to when comparing across various optimizers and hyperparameters. Regarding your comment “...I didn’t feel I gained a clear insight to help decide on a better optimizer…”, this actually touches on the primary takeaway from our ablations: across the optimizers we tested, there isn’t any clear winner based on both optimal performance and hyperparameter stability. The only differing axis for practitioners to consider is thus memory constraints, but based on our results, they can be reassured that memory considerations for optimizer choice should not significantly affect the performance of their training run, at least across the optimizers tested (i.e. optimizers using diagonal preconditioning). For more concrete takeaways regarding our hyperparameter sweeps, we reported the optimal learning rates found in our sweeps in Appendix A as a starting point for practitioners in tuning (eg. tuning optimizers like Signum and Lion seem to require learning rates an order of magnitude smaller compared to Adam/Adafactor). In terms of other tuning practices, we observed most other parameters are very stable for the optimal learning rate (with exception to momentum) and thus we believe tuning learning rate and momentum are most critical. We agree that the main takeaways from our results could have been consolidated and communicated more clearly; we will be sure to add everything that we have written above in a paragraph in the Discussion section in the revised pdf (please also see our response to Reviewer 4oK2 for a similar discussion). Regarding your last point (“...the paper has potential but feels incomplete”), if there’s anything in addition to our clarifications above that could provide a more complete message to readers, we would appreciate hearing any concrete suggestions and would look into implementing those.
>
> [1] Kunstner, Frederik, et al. "Heavy-tailed class imbalance and why adam outperforms gradient descent on language models." (2024).
>
> [2] Zohrevand, Abbas, and Zahra Imani. "An empirical study of the performance of different optimizers in the deep neural networks." 2022 International Conference on Machine Vision and Image Processing (MVIP). IEEE, 2022.
>
> [3] Wortsman, Mitchell, et al. "Small-scale proxies for large-scale transformer training instabilities."(2023).

---

> > ### Author Response · Authors · 2024-11-19
> > **Response to Reviewer (2/2)**
> >
> > ### Questions:
> > 1. **Regarding potential N-dimensional interactions**: Indeed, the problem of finding the hyperparameter combination which obtains the ‘optimal’ solution is an N-dimensional one; in line 165, when we write ‘2D’ interactions, we refer to any higher dimensional interactions that aren’t covered by our 1-dimensional sweeps after tuning for learning rate (we will make this more clear in the revision). We acknowledge this being a limitation of our work and ideally if we weren’t compute-constrained we could explore higher-dimensional interactions between hyperparameters. However, we still believe our results can be insightful to practitioners by providing a signal about hyperparameter stability in a region around parameters that are common in practice. Most practitioners are also unable to explore the full search space and usually adopt hyperparameters from previous work using similar architectures in similar settings; our results indicate that one tractable methodology for identifying the ‘optimal’ setting in this region is sweeping across the slice of learning rate and momentum, which were the only two hyperparameters that had significant effects on performance.
> > 2. **Regarding using Bayesian optimization**: We agree that if the goal was just to find optimal hyperparameters, that things like Bayesian optimization could work well. However, the main goal of our paper is not to do hyperparameter optimization, it is to scientifically understand the sensitivity of various hyperparameters as a way to better understand the relative merits of different choices when constructing optimizers. This is why we propose the 1D sweep methodology which allows us to conduct these rigorous ablations at reasonable computational scale.
> >
> > We hope that we adequately addressed the reviewer’s concerns and questions, and are happy to provide any additional clarifications and engage in further discussion.

---

> > > ### Author Response · Authors · 2024-11-24
> > >
> > > We would like to thank the reviewer again for their valuable feedback and suggestions. We would be grateful if the reviewer would consider updating their score based on our response. If not, we would be happy to answer any additional questions.

---

### Official Review · Reviewer_9bTN · 2024-11-10

**Soundness:** 3
**Presentation:** 4
**Contribution:** 3
**Rating:** 8
**Confidence:** 4

**Summary:**

This paper conducts a thorough experimental evaluation of the performance and stability of common optimizers used in pre-training language models and uncovers interesting and previously unknown parameters in the transformer architecture that seemingly benefit more from the adaptivity of some of these optimizers.

**Strengths:**

* The thorough empirical evaluation fulfills a missing piece in the current literature of optimizers used in training language models, as most works in this literature stick to Adam.
* The experimental grids used for different experiments seem sufficient to draw robust conclusions. The authors incorporate models of different scales and explore 1 dimensional grid searches which is reasonable given the intense compute needs of such studies.
* This work uncovers an interesting phenomenon which attributes most of the adaptivity gains from Adam and similar algorithms to normalization and last layer's parameters.
* Considering stability of the training algorithms wrt hyperaparameters as opposed to the final performance is an important aspect since eventually a practitioner wouldn't be able to find the best possible hyperparameters but the hope would be to be able to easily food "good" set of hyperparameters.

**Weaknesses:**

* One major drawback of this study could be the lack of a search over min learning rate at the end of cosine scheduling. As the authors have mentioned, learning rate plays the most important role among all these parameters. Recent studies (https://arxiv.org/abs/2410.05192) have shown that the final stage of training where the learning rate is decreased down to the minimum learning rate plays a crucial role in determining the final performance. Based on this, I'd be curious to know if certain algorithms would outperform others if the min learning rates were chosen carefully, since in that case individual momentums and how the history of gradients is dealt with could significantly differ among different algorithms. I would predict that using some optimizers, some larger learning rates would result in drastically better performances given that the minimum learning rate is small enough.

* I think implementing AdaFactor and AdaLayer would be more fruitful if parameters corresponding to different attention heads would be disentangled. Since the authors don't mention anything in this regard, I assume that they haven't implemented these algorithms to do so, but the nature of computations in attention heads would maybe benefit from adaptivity isolated to the attention head over all of the attention parameters.

* Recently shampoo has gained significant popularity in training language models. It would be nice if the authors could incorporate shampoo among these optimizers to have a more complete story.

* It would be nice if the authors could elaborate on how optimality of different hyperparameters changes as the scale increases.

**Questions:**

* It is intuitive to me that adaptivity can help with optimization of LayerNorm as the affine parameters are applied in a coordinate-wise way, but I can't understand why the last layer (which I think is the embedding layer in the case of these models?) needs adaptivity. Can the authors please elaborate on this?

---

> ### Author Response · Authors · 2024-11-19
>
> Thanks for the insightful comments. We are happy to see that the reviewer finds our work interesting, and appreciates the focus on hyperparameter stability. The specific comments are addressed below:
> 1. **Sweeping min cosine learning rate** - We did sweep this hyperparameter with Adam and used 0.1 as it seemed to perform the best for Adam in our setup. We will explicitly add the sweeps for this parameter in future revisions.
> 2. **Attention heads disentangling in Adafactor and Adalayer** - Indeed, from the Hessian structure perspective, Adam-mini [1] suggests having an adaptive learning rate assigned to each attention head’s query and key matrix could improve Adalayer* performance further. This is an interesting direction of future work and we would be happy to add these variations in a future revision.
> 3. **Shampoo** - We explicitly wanted to limit our study to the diagonal preconditioning optimizers. This is the reason we have not added Shampoo to the study.
> 4. **Optimality of hyperparameters with scale** - Most of our sweeps were done at 150m scale,and thus we cannot provide the scaling of optimal hyperparameters with scale given our compute budget. Regarding learning rate, keeping other hyperparameters fixed, we found the optimal learning rate to be reasonably stable from 150m to 1.2b scale (as shown in Table 1 in the Appendix).
> 5. **Last layer adaptivity** - This is required as different tokens have varying frequencies in the training data, thus, without adaptivity, the infrequent tokens won’t even be trained. For this reason, we need adaptivity in the last layer.
>
> We hope that we adequately addressed the reviewer’s concerns and questions, and are happy to provide any additional clarifications or engage in further discussion.
>
> [1] Zhang, Yushun, et al. "Adam-mini: Use fewer learning rates to gain more." (2024).

---

### Official Review · Reviewer_J9Wq · 2024-11-10

**Soundness:** 3
**Presentation:** 3
**Contribution:** 2
**Rating:** 6
**Confidence:** 3

**Summary:**

The study compares optimization algorithms, including SGD, Adafactor, Adam, Lion, and Sophia, for training language models of different sizes and architectures. Results show that, apart from SGD, all algorithms achieve similar performance and robustness to hyperparameter variations, suggesting practical considerations like memory and simplicity may guide optimizer choice. The researchers further dissect Signum and Adalayer to explore Adam’s adaptivity effects. They find that adaptivity on the last layer and LayerNorm parameters is essential for maintaining performance and stability, highlighting these elements as crucial for reliable training.

**Strengths:**

1. This paper is well-written and easy to understand. The experimental setups, discussion and interpretations of the results, and limitations of the methods are spelled out clearly. Though it might not have great novelty, I think the extensive study on hyperparameters on different optimizers is a valuable contribution to the LLM optimization community.  I enjoyed reading this paper.
2. Soundness of the result: As the authors mentioned in the paper, due to limits of computational resources, they only do 1D hyperparameter sweep, instead of 2D or even higher-order sweeps. This is fine for the major finding of the paper, which is a positive result, that the non-SGD mainstread optimizers are robust to choice of wide range of hyperparameters. Tuning remaining hyperparameters will definitely make the best performance of the current hyperparameter better.
3. The finding that last layer and layernorm parameters need more  fine-grained (than layerwise) adaptivity is interesting. It shed lights on future research towards how Adam works and memory-efficient variant of Adam.

=============
I increased my score to 6 in reflection of the authors' additional experiments and clarifications.

**Weaknesses:**

Though I appreciate the extensive abalation on the LLM training experiments with different hyperparameters, the interpretation and the takeways of the experiments are vague and imprecise. I will elaborate it below.

1. The authors often use "Comparable" to describe the performance of models trained with different hyperparameters, which I could not find a precise definition in the paper. A validation loss of 3.07 may look close to 3.10 in terms of the relative ratio, but because the minimum of loss is not zero, the suboptimality between 3.07 and 3.10 maybe very significant, say if the minimum of population loss is 3.0 for 150M models.

    As a result of imprecise definition, it is not clear what is the consequence of having comparable validation loss. Does it imply such gap between comparable losses are negligible so in practice if people use any hyper-parameters which lead to losses comparable to optimal validation loss in the first attempt, they are satisfied and do not need to spend more compute to rerun the experiments for the optimal optimal validation loss? If they would still like to rerun the experiment, no matter how small the gap in validation loss is, having a wide range of hyperparameters lead to comparable performance does not imply ease of hyperparameter tuning.

2. The authors write in line 327 "Takeaway: generally algorithms are more stable with respect to other hyperparameters and the possible gains in performance are relatively small compared to learning rate and momentum."

    I do not fully agree with this claim take-home message, given the experiments in the current draft. It seems that the authors get this conclusion just from the shape of the loss curves in those 1D hyperparameter sweep plots, instead of based on some quantative metrics. This could be problematic, because the range of some hyperparameter sweep in section 3.5 for hyperparameters other than learning rate and momentum seems to be smaller than that of learning rate and momentum. I am convinced that for sufficiently small WD and epsilon, the final validation losses are nearly the same. But for warmup, batch size, and $beta_2$, the range are smaller and thus I am not convinced. For example, the ratio between maximal and minimal batch sizes tried in the experiments are just 8, while the ratio is more than a thousand for learning rate. But given that the authors are training in the "chinchilla optimal" regime, which means the flops are fixed for different batch sizes, we should expect changing batch size can have similar effect to changing learning rates given the linear scaling rule for SGD [Goyal et al., 2017, Li er al.,2021] or square-root scaling rules for Adam [Malladi et al., 2022]. Therefore I suspect if the authors also vary batch size in a range of 1000 times, the impact of batch size would be much larger. (The authors also mention the 2D effect between batch size and learning rate in line 166)

    For pretraining percentage and $beta_2$, I encourage the authors to include more extreme values to support the claim that these hyperparameters do not matter. Instead showing them they do not matter in the current small range, it is more informative to show to the readers at what extreme values the loss starts to increase significantly. It is possible for warmup that we can completely get rid of it, and in this case it is useful to include that in the plot, just like weight decay.


**Soundness of finding in Section 3.6**: First the citation of Lemma 1 from [Balles\&Henni, 2018] seems to be quite different from what that is in the cited paper. Second, given the assumption in the Lemma 1, $\delta_{adam}$ seems just defined as $\frac{\mathbb{E}[m]}{\sqrt{\mathbb{E}[v^2]}}$, and the lemma implies that $\delta_{signum}$ must be $1$, which is not necessarily true, when sign of $m$ is negative. Maybe the authors are missing an absolute sign on $\mathbb{E}[m]$.

I also do not understand the argument between line 357-line 360 when $\beta_1=\beta_2$. The authors are essentially claiming for all coordinates and all time steps, the first and square root of second moment have the same ratio, which I do not see why it should be true. This prediction could be easily checked by just plotting the update magnitude per coordinate for Adam and see if they are roughly the same. If the authors allow a different scaling factor for every coordinates, then this claim becomes trivial because all first-order optimization algorithm using diagonal preconditioning have the same sign for each coordinate in its update and only differ by scaling.

**Questions:**

Despite the issues mentioned in the weakness, I have the following additional questions:

1. why the loss curve for signum are different in Figure 4, $\beta_1$ and Figure 5, $\beta_2$? Signum is defined as lion with $\beta_1=\beta_2$, if I remember correctly.

2. It would be interesting to add AdaSGD (Wang et al., 2020) into comparison, which only uses one more register than normal SGD, but has much better convergence, stability and tolerence to larger learning rate. AdaSGD can be viewed as a global version of adalayer, i.e., by viewing all the parameters as one layer. A very recent work by [Xie et al.,2024] shows that AdaSGD closes a large portion of the gap between SGD and Adam on GPT2. Replacing the SGD part in adalayer by AdaSGD might help us better understand the importance of adaptivity at different levels of granularity, under the unified framework of AdaLayer, or more generally Adam for arbitrary partition of parameter blocks.

3. typo. "in perplexity" in line 317 should be "in validation loss"?

I am willing to improve my score if the authors could address my concerns.

---

> ### Author Response · Authors · 2024-11-19
>
> We thank the reviewer for their insightful comments. We are happy to see that the reviewer finds the paper well-written and easy to understand. The specific comments are addressed below:
>
> 1. **Comparable performance of optimizers** - As discussed in the introduction, the primary motivation for our initial experiments was to empirically verify the community’s preconceptions about Adam being superior to other optimizers; this implies an underlying assumption of a substantial gap existing between Adam and these other optimizers. When we have written these optimizers are "comparable in performance," we mean that the actual performance differences between them is much smaller than what might have previously been perceived, especially when we consider this gap relative to the gap of the outlier performance of SGD.  As seen in Figure 1, compared to SGD, other algorithms are relatively stable when tuning learning rate. This does not mean that practitioners should not tune learning rate as a hyperparameter, but our intended main takeaway is that at scale–where it may only be possible to launch a single run– optimizers apart from SGD will not be very far from their optimal performance. This is an essential property that an optimizer needs to satisfy to be used in large-scale pre-training runs. This is indeed a comparison on a more crude scale, but we believe it is justified to refrain from defining a precise quantitative gap because such a definition may not be robust to other design choices in our setup, such as architectural variations. If the reviewer disagrees with our reasoning, we are keen to hear their thoughts to discuss this point further.
> 2. **Stability of other hyperparameters except momentum and learning rate** - Yes, we agree that our takeaway in line 327 is based on the relative ‘visual’ stability of the 1D plots. We would be happy to make it quantitative by providing the range for each hyperparameter which incurs a maximum deviation of a specified quantity (may be 0.01) from the optimal along the 1D sweep. If the reviewer considers this information important for their evaluation of the paper, please make note and we would be glad to include it.
> 3. **Sweep of $\beta_2$ and warmup** - We would be happy to add more extreme values for the two parameters in the sweeps and will include this in the revision.
> 4. **Sweep of batch size** - As pointed out by the reviewer, when changing batch size, tuning learning rate and momentum is known to be essential for obtaining optimal performance. This was the reason we did not do large sweeps for this hyperparameter. We would be happy to add more points in the 1-D sweep for batch size as well.
> 5. **Section 3.6 finding** - We are sorry for the confusion. We did not mean to refer to Lemma 1 of Balles and Hannig 2018, and simply wanted to point out that this observation has been noted in this previous work. Also, thank you for the correction– we indeed need |E[m]| in the numerator. Regarding the discussion when $\beta_1 = \beta_2$, we only mean to say that as both the numerator and denominator average over the same window size, it is more likely for Adam and Signum to behave similarly. Indeed, for this to hold mathematically, we will need the ratio of the mean to the square root of the second moment to be similar for all the coordinates, which is unlikely to happen. However, as shown in Figure 6, we do get similar performance as Signum for Adam with $\beta_1 = \beta_2$.
> 6. **Loss curve for Signum in Figure 4 and Figure 5** - Indeed, in the $\beta_2$ sweep in Figure 5, we don’t have a curve for Signum, and that is precisely because Signum is Lion with $\beta_1 = \beta_2$.
> 7. **AdaSGD** - We would be very happy to include AdaSGD in subsequent revisions.
> 8. **Typos** - Thanks for catching them. We will fix them.
>
> We hope that we adequately addressed the reviewer’s concerns and questions, and are happy to provide any additional clarifications or engage in further discussion.

---

> > ### Author Response · Authors · 2024-11-24
> > **AdaSGD**
> >
> > We have performed sweeps over AdaSGD learning rate as well as replacing SGD with AdaSGD for the SGD + Adalayer* experiments, following the implementation from [Xie et al, 2024]. When sweeping across learning rate (see figure here: https://postimg.cc/87zRyDc4) we observed that AdaSGD performs marginally better than SGD but still lacks learning rate stability; this aligns with our previous findings in Figure 7, where we saw that certain parameters had effective learning rates which were multiple orders of magnitude different from other parameters, and thus a single adaptive learning rate across all parameters likely would not suffice for achieving performance and stability. When replacing SGD with AdaSGD on the matrix parameters and using Adalayer* to train the last layer and LayerNorm parameters (see figure here: https://postimg.cc/SJrL7jSj) the performance and stability has improved to essentially match that of Adalayer*, but a gap still remains between Adam performance even with an adaptive learning rate across matrix parameters; it is likely that this small gap is attributed to the need for increased adaptivity granularity for certain matrix parameters (eg. across attention heads, as Reviewer 9bTN suggested). However, we reiterate that a large fraction of the gap between SGD/AdaSGD performance and Adam performance was closed by adding adaptivity to the LayerNorm and last layer parameters, despite comprising only a small fraction of the total parameters.
> >
> > We would like to thank the reviewer again for their valuable feedback and suggestions. We would be grateful if the reviewer would consider updating their score based on our response. If not, we would be happy to answer any additional questions.

---

> > > ### Comment · Reviewer_J9Wq · 2024-11-24
> > >
> > > I thank the authors for their clarification and additional experiments.
> > >
> > > 1. Now I better understand the motivation of this study, though I personally do not find it very surprising to show adam is not significantly or consistently better than lion.
> > >
> > > 2. I appreciate the additional experiments on AdaSGD and found it quite interesting that AdaLayer* + AdaSGD slightly beats AdaLayer*. This could shed light on future research on effectiveness of Adam and suggest the authors to include the related experiments into next revision.
> > >
> > > 3. Given the additional parameter sweep over batch size, I think the authors should update the takeaway in line 295 of the updated draft and include batch size into those parameters that performance are less stable with, similar to learning rates and momentum.
> > >
> > > 4. Given authors' explanation, I think the reasoning in section 3.6 should be removed, or at least it should be written in a less misleading way. The empirical observation that when $\beta_1=\beta_2$, adam is close to signum is still valuable, but the theoretical derivation is only a heuristic/motivation to run the proposed experiments, rather than something holds either empirically or mathematically.
> > >
> > > I will increase my score to 6 If the authors agree with edits (2),(3),(4) mentioned above. Otherwise I would like to hear authors' thoughts/reasoning against these.

---

> > > > ### Author Response · Authors · 2024-11-24
> > > >
> > > > We are happy to incorporate all three suggestions in the next revision. We thank the reviewer for engaging with our work and follow-up responses.

---

### Author Response · Authors · 2024-11-24

We would like to thank all reviewers for taking the time to engage with our work and provide insightful feedback. Before the discussion period ends, we’d like to address some shared comments below and highlight revisions that we have made to the pdf in light of reviewer suggestions.

**Regarding the contribution of our work and main takeaways:** We would like to reiterate the main contributions of our work and highlight their broader implications. Our paper provides a comprehensive empirical evaluation of optimizers for autoregressive language model pretraining, including a wide range of hyperparameters and model scales. In previous work, as we have noted in the manuscript, empirical studies of optimizers across hyperparameters have been either outside of the autoregressive language modeling domain, limited to Adam only, or limited to a fixed scale. One key finding is that no single algorithm consistently outperforms others in terms of both performance and hyperparameter stability, challenging the prevailing notion of Adam as the default optimizer for language models. Additionally, we show that adaptivity in both the last layer and LayerNorm parameters is essential for maintaining performance and learning rate stability. This insight not only informs optimization design but also suggests pathways for developing more memory-efficient optimizers—a direction that concurrent work has already started to explore. We believe our findings can encourage groups with more computational resources or those working in other domains to replicate and extend our results, scaling to larger models and exploring new designs. We agree that the main takeaways from our results could have been consolidated and communicated more clearly, and we have revised the pdf to include a much more detailed discussion of this (see below).

We have also uploaded an updated pdf, with specific changes listed below:
* We have updated Figure 5 (see also linked here: https://postimg.cc/yDpFt5SH) to include more extreme values for different hyperparameters, namely batch size, warmup, and beta_2. We note two observations in light of the expanded sweeps: Lion shows poorer performance at extreme values of $\beta_2$. However, it is important to note that the $\beta_2$ parameter in Lion functions more like a form of ``momentum", whereas in Adam and Adafactor, $\beta_2$ regulates the moving average of the squared gradients. We also see that the performance across optimizers worsens at high batch size. This is expected as optimization algorithms are known to exhibit diminishing performance with increasing batch size.
* We have updated the introduction to highlight the aim of our work in performing rigorous benchmarking to validate the community’s preconception of Adam’s performance relative to other diagonal preconditioning optimizers, and explore further the need for adaptivity in different parts of the network.
* We have updated the discussion section with two paragraphs that comprehensively summarizes the main takeaways of the results from our work (from the initial sweeps and the follow-up investigation on the need for adaptivity on different network parameters).
* We have moved Related Works to the Appendix and added a paragraph highlighting work on benchmarking machine learning optimizers.
* We have improved the visibility and font size for a few plots (Figure 8 in particular), and addressed some typos brought up by reviewers.

Before the discussion period ends, we would be very grateful if the reviewers would consider updating their score based on our responses. If not, we would be happy to answer any additional questions.

---

### Meta-Review · Area_Chair_LgGn · 2024-12-20

**Metareview:**

This paper is an empirical paper that studies how different optimizers perform for optimizing transformers in the context of LLMs. It comprehensively benchmarked multiple optimizers and exhibit some insights about their performance and also studied several new variants. The results of the paper may provide guidance about which optimizer to use for training LLMs given their constraints on memory, etc. While some reviewers criticize that the paper does not propose new algorithms, its comprehensive comparison between preconditioning based optimizers and its insights on adaptivity of the last layer are appreciated by reviewers. The paper could inspire more work to improve the training of transformers. Given the overall positive ratings of the paper, I recommend an acceptance.

**Additional Comments On Reviewer Discussion:**

The authors have provided detailed rebuttal to address reviewers' questions and concerns. Some reviewers have involved in the discussion with reviewers and acknowledge their efforts. Others do not respond to rebuttal.

---

### Decision · Program_Chairs · 2025-01-22

Accept (Poster)